# Image Quality Assessment: Integrating Model-Centric and Data-Centric Approaches

Peibei Cao[1], Dingquan Li[2], and Kede Ma[1]

[1]City University of Hong Kong, [2]Peng Cheng Laboratory

peibeicao2-c@my.cityu.edu.hk, dingquanli@pku.edu.cn, kede.ma@cityu.edu.hk

Learning-based image quality assessment (IQA) has made remarkable progress in the past decade, but nearly all consider the two key components—model and data—in isolation. Specifically, model-centric IQA focuses on developing "better" objective quality methods on fixed and extensively reused datasets, with a great danger of overfitting. Data-centric IQA involves conducting psychophysical experiments to construct "better" human-annotated datasets, which unfortunately ignores current IQA models during dataset creation. In this paper, we first design a series of experiments to probe computationally that such isolation of model and data impedes further progress of IQA. We then describe a computational framework that integrates model-centric and data-centric IQA. As a specific example, we design computational modules to quantify the sampling-worthiness of candidate images. Experimental results show that the proposed sampling-worthiness module successfully spots diverse failures of the examined blind IQA models, which are indeed worthy samples to be included in next-generation datasets.

## 1. Introduction

Image quality assessment (IQA) is indispensable in a broad range of image processing and computational vision applications. A learning-based IQA system [1–3] generally has two key components: the engine "model" and its fuel "data." The system is learned to predict image quality from a large number of human-annotated data. From this perspective, it is natural to categorize IQA studies into model-centric and data-centric approaches.

The goal of model-centric IQA [4–8] is to build computational methods that provide consistent predictions with human perception of image quality. Improved learning-based IQA models have been developed from the aspects of computational structures, objective functions, and optimization techniques. Particularly, the *computational structures* have shifted from shallow [9] to deep methods with cascaded linear and nonlinear stages [10]. Effective quality computation operators have also been identified along the way, such as generalized divisive normalization (GDN) over half-wave rectification (ReLU) [11], bilinear pooling over global average pooling [12], and adaptive convolution over standard convolution [7]. The *objective functions* mainly pertain to the formulation of IQA. It is intuitive to think of visual quality as absolute quantity and employ the Minkowski metric to measure the prediction error. Another popular learning-to-rank formulation [13] treats perceptual quality as relative quantity, which admits a family of pairwise and listwise ranking losses [3, 14]. Other loss functions for accelerated convergence [15] and uncertainty quantification [8, 16], have also begun to emerge. The *optimization techniques* in IQA benefit significantly from practical tricks to train large-scale convolutional neural networks (CNNs) for visual recognition [17]. One learning strategy specific to IQA is the ranking-based dataset combination trick [8], which enables an IQA model to be trained on multiple datasets without perceptual scale realignment.

The goal of data-centric IQA is to construct human-rated IQA datasets via psychophysical experiments for the purpose of benchmarking and developing objective IQA models. A common theme in data-centric IQA [18–21] is to design efficient subjective testing methodologies to collect reliable human ratings of image quality, typically in the form of mean opinion scores (MOSs). Extensive practice [22] seems to show that there is no free lunch in data-centric IQA: collecting more reliable MOSs generally requires more delicate and time-consuming psychophysical procedures, such as adopting the two-alternative forced choice (2AFC) method in a well-controlled laboratory environment with proper instructions. Bayesian experimental designs have also been implemented [23, 24] in an attempt to improve rating efficiency. Arguably, a more crucial step in data-centric IQA is

First Conference on Parsimony and Learning (CPAL 2024).

sample selection, which is, however, much under-studied. Vonikakis *et al.* [25] proposed a dataset shaping technique to identify the image subset with uniformly distributed attributes of interest. Cao *et al.* [26] described a sample selection method in the context of real-world image enhancement based on the principle of maximum discrepancy competition [27, 28].

Although the past achievements in IQA are worth celebrating, only weak connections have been made between model-centric and data-centric IQA, which we argue is the primary impediment to further progress of IQA (see Figure 1). From the model perspective, objective methods are optimized and evaluated on fixed (and extensively reused) sets of data, leading to the *overfitting* problem. From the data perspective, IQA datasets are generally constructed while being blind to existing objective models. This may cause the *easy dataset* problem [21]: the newly created datasets expose few failures of existing IQA models, resulting in a significant waste of the expensive human labeling budget.

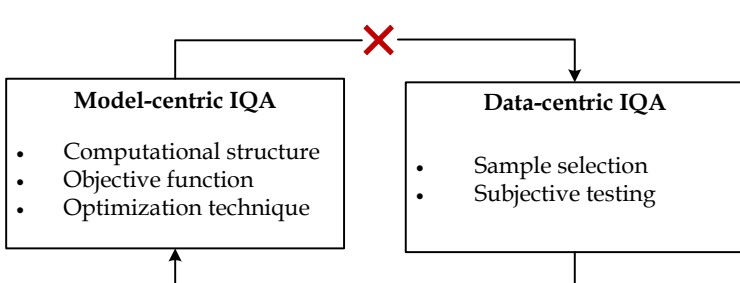

Figure 1: Past work makes weak connections between data-centric and model-centric IQA, hindering the further progress of the field. The connection from data-centric IQA to model-centric IQA embodies a tremendous amount of work on how to train "accurate" IQA models on human-rated datasets. The missing part for closing the loop is to leverage model-centric IQA to guide the design of data-centric IQA, especially in sample selection.

In this paper, we take initial steps towards integrating model-centric and data-centric IQA. Our main contributions are threefold.

- We design a series of experiments to probe computationally the overfitting problem and the easy dataset problem of blind IQA (BIQA) in real settings.

- We describe a computational framework that integrates model-centric and data-centric IQA. The key idea is to augment the (main) quality predictor with an auxiliary computational module to score the sampling-worthiness of candidate images for dataset construction.

- We provide a specific example of this framework, where we start with a (fixed) "top-performing" BIQA model, and train a failure predictor by learning to rank its prediction errors. Our sampling-worthiness module is the weighted combination of the learned failure predictor and a diversity measure computed as the semantic distance between deep content-aware features [29]. Experiments show that the proposed sampling-worthiness module is able to spot diverse failures of existing BIQA models in comparison to several deep active learning methods [30–33]. These samples are indeed worthy of being incorporated into next-generation IQA datasets.

## 2. Related Work

**Model-Centric IQA**. We will focus on reviewing model-centric BIQA, which improves quality prediction from the model perspective without reliance on original undistorted images. Conventional BIQA models pre-defined non-learnable *computational structures* to extract natural scene statistics (NSS). Learning occurred at the quality regression stage by fitting a mapping from NSS to MOSs. Commonly, NSS were extracted in the transform domain [34, 35], where the statistical irregularities can be more easily characterized. Nevertheless, transform-based methods were slow, which motivated BIQA models in the spatial domain [1, 4].

With the latest advances in CNNs, learnable computational structures have revolutionized the field of BIQA [5–8]. Along this line, the *objective functions* are important in guiding the optimization of BIQA models. It is straightforward to formulate BIQA as regression, and the Minkowski metric is the objective function of choice. An alternative view of BIQA is through the lens of learning-to-rank [14], with the goal of inferring relative rather than absolute quality. Pairwise and list-wise learning-to-rank objectives have been successfully adopted. One last ingredient of model-

centric IQA is the selection of *optimization techniques* for effective model training, especially when the human-rated perceptual data are scarce. Fine-tuning from pre-trained CNNs on other vision tasks [6], patchwise training [36], and quality-aware pre-training [7] are practical optimization tricks in BIQA. Of particular interest is the unified optimization strategy by Zhang *et al.* [8], allowing a single model to learn from multiple datasets simultaneously.

**Data-Centric IQA** improves the quality prediction performance from the data perspective, which consists of two major steps: sample selection and subjective testing. The immediate output is a human-rated dataset for training and benchmarking objective IQA models. For a long time, the research focus of data-centric IQA has been designing reliable and efficient *subjective testing* methodologies for MOS collection [37, 38]. It is generally believed that the 2AFC design in a well-controlled laboratory environment is more reliable than single-stimulus and multiple-stimulus methods. However, the cost to exhaust all paired comparisons is prohibitively expensive when the image size is large.

*Sample selection* is perhaps more crucial in data-centric IQA but experiences much less success. Early datasets, like LIVE [18], TID2008 [39], and CSIQ [40], selected images with simulated distortions. Due to the combination of image content, distortion type and level, the number of distinct reference images is often limited. Recent large-scale IQA datasets began to contain images with realistic camera distortions, including CLIVE [20], KonIQ-10k [10], SPAQ [21], and PaQ-2-PiQ [41]. Such a shift in sample selection provides a good test for synthetic-to-realistic generalization.

Sample diversity during dataset creation has also been taken into account. Vonikakiset *et al.* [25] cast sample diversity as a mixed integer linear programming, and selected a dataset with uniformly distributed image attributes. Euclidean distances between deep features [10, 26] are also used to measure semantic similarity. Nevertheless, sample diversity is only one piece of the story; often, the included images are too easy to challenge existing IQA models (see Sec. 3.2).

**UNIQUE** [8] is a recently proposed BIQA model, which combines multiple IQA datasets as the training data. Specifically, assuming Gaussianity of the true perceptual quality $q(x)$ with mean $\mu(x)$ and variance $\sigma^2(x)$ and the independence of quality variability across images, the probability of image $x$ having higher perceptual quality than image $y$ can be calculated as

$$p(x, y) = \Pr(q(x) \geq q(y)) = \Phi\left(\frac{\mu(x) - \mu(y)}{\sqrt{\sigma^2(x) + \sigma^2(y)}}\right), \qquad (1)$$

where $\Phi(\cdot)$ denotes the standard Gaussian cumulative distribution function. From the $i$-th dataset for a total of $N$ datasets, UNIQUE randomly samples $N_i$ pairs of images $\{(x_j^{(i)}, y_j^{(i)})\}_{j=1}^{N_i}$, and the combined training dataset is thus in the form of $\mathcal{D} = \{\{(x_j^{(i)}, y_j^{(i)}), p_j^{(i)}\}_{j=1}^{N_i}\}_{i=1}^{N}$.

UNIQUE aims to learn two differentiable functions $f_w(\cdot)$ and $\sigma_w(\cdot)$, parameterized by a vector $w$, for quality and uncertainty estimation. The prediction of $p(x, y)$ can be done by replacing $\mu(\cdot)$ and $\sigma(\cdot)$ with their respective estimates:

$$\hat{p}(x, y) = \Phi\left(\frac{f_w(x) - f_w(y)}{\sqrt{\sigma_w^2(x) + \sigma_w^2(y)}}\right). \qquad (2)$$

UNIQUE minimizes the fidelity loss [42] between the two probability distributions $p(x, y)$ and $\hat{p}(x, y)$ for parameter optimization:

$$\ell(x, y, p) = 1 - \sqrt{p(x, y)\hat{p}(x, y)} - \sqrt{(1 - p(x, y))(1 - \hat{p}(x, y))}. \qquad (3)$$

To resolve the scaling ambiguity in Eq. (2) and to resemble the human uncertainty when perceiving digital images, UNIQUE adds a hinge-like regularizer during training [8]. The original UNIQUE employs ResNet-34 [43] as the backbone followed by bilinear pooling and $\ell_2$-normalization, and implements $f_w(\cdot)$ and $\sigma_w(\cdot)$ as the two outputs of a fully connected (FC) layer. We will adopt UNIQUE in our subsequent experiments.

# 3. Probing Problems in the Progress of BIQA

## 3.1. Overfitting Problem

As there is no standardized computable definition of overfitting, especially in the context of deep learning [44], quantifying overfitting is still a wide open problem. Here, we choose to use the group maximum differentiation (gMAD) competition [45] to probe the generalization of BIQA models to a large-scale unlabeled dataset. gMAD is a discrete instantiation of the MAD competition [27] that relies on synthesized images to optimally distinguish the models. As opposed to the *average-case* performance measured on existing IQA datasets, say by Spearman's rank correlation coefficient (SRCC), gMAD can be seen as a *worst-case* performance test by comparing the models using extremal image pairs that are likely to falsify them. We declare an overfitting case of a BIQA model if it shows strong average performance on standard IQA datasets but weak performance in the gMAD competition.

**Experimental Setup**. We choose nine BIQA models from 2017 to 2021: RankIQA [5], DeepIQA [36], NIMA [6], KonCept512 [10], Fang2020 [21], HyperIQA [7], LinearityIQA [15], UNIQUE [8], and MetaIQA+ [46]. Table 1 shows the SRCC results between the algorithm publishing time and the performance ranking[1] on four widely used IQA datasets, BID [19],

Table 1: SRCC between the performance ranking of nine BIQA methods and their publishing time on four datasets. It is clear that the quality prediction performance "improves" steadily over time.

| Dataset | BID [19] | CLIVE [20] | KonIQ-10k [20] | SPAQ [21] |
|---------|----------|------------|----------------|-----------|
| SRCC | 0.6946 | 0.7950 | 0.6695 | 0.7029 |

CLIVE [20], KonIQ-10k [10], and SPAQ [21]. We find that "steady progress" over the years has been made by employing more complicated computational structures and advanced optimization techniques.

We now set the stage for the nine BIQA models to perform the gMAD competition. Specifically, we first gather a large-scale unlabeled dataset $\mathcal{U}$, containing 100,000 photographic images with marginal distributions nearly uniform w.r.t. five image attributes (*i.e.*, JPEG compression ratio, brightness, colorfulness, contrast, and sharpness). Our dataset covers a wide range of realistic camera distortions, such as sensor noise contamination, motion and out-of-focus blur, under/over-exposure, contrast reduction, color cast, and a mixture of them. Given two BIQA models $f_i(\cdot)$ and $f_j(\cdot)$, gMAD [45] selects top-$K$ image pairs that best discriminate between them:

$$(x_k^\star, y_k^\star) = \arg\max_{x,y} f_i(x) - f_i(y), \text{s.t. } f_j(x) = f_j(y) = \alpha, \ x, y \in \mathcal{U} \setminus \mathcal{D}_{k-1}, \tag{4}$$

where $\mathcal{D}_{k-1} = \{x_{k'}^\star, y_{k'}^\star\}_{k'=1}^{k-1}$ is the current gMAD image set. The $k$-th image pair must lie on the $\alpha$-level set of $f_j$, where $\alpha$ specifies a quality level. The roles of $f_i$ and $f_j$ should be switched. $Q$ (non-overlapping) quality levels are selected to cover the full quality spectrum. By exhausting all distinct pairs of BIQA models and quality levels, we arrive at a gMAD set $\mathcal{D}$ that contains a total of $9 \times 8 \times 5 \times 2 = 720$ image pairs, where we set $Q = 5$ and $K = 2$.

We invite 25 human subjects (12 males and 13 females) to gather perceived quality judgments of each gMAD pair using the 2AFC method. They are forced to choose the image with higher perceived quality for the 720 paired comparisons. The 25 subjects are mostly young researchers (aged between 22 and 30) with a computer science background but are unaware of the goal of this work. They are asked to finish the experiments in an office environment with normal lighting conditions and without reflecting ceiling walls and floors.

After subjective testing, we obtain the raw pairwise comparison matrix $A \in \mathbb{R}^{9 \times 9}$, where $a_{ij} \in \{0, 1 \ldots, 250\}$ indicates the counts of $x^\star$ preferred over $y^\star$ by the 25 subjects on the ten associated image pairs (by solving Problem (4)). We compute, from $A$, a second matrix $B \in \mathbb{R}^{9 \times 9}$, where $b_{ij} = a_{ij}/a_{ji}$ denotes the pairwise dominance of $f_i$ over $f_j$. Laplace smoothing [47] is applied when $a_{ji}$ is close to zero. We convert the pairwise comparisons into a global ranking $r \in \mathbb{R}^9$ using Perron rank [48]. A larger $r_i$ indicates better performance of $f_i$ in the gMAD competition.

---

[1]As each BIQA model assumes different (and unknown) training and testing splits, for a less biased comparison, we compute the quality prediction performance on the full dataset.

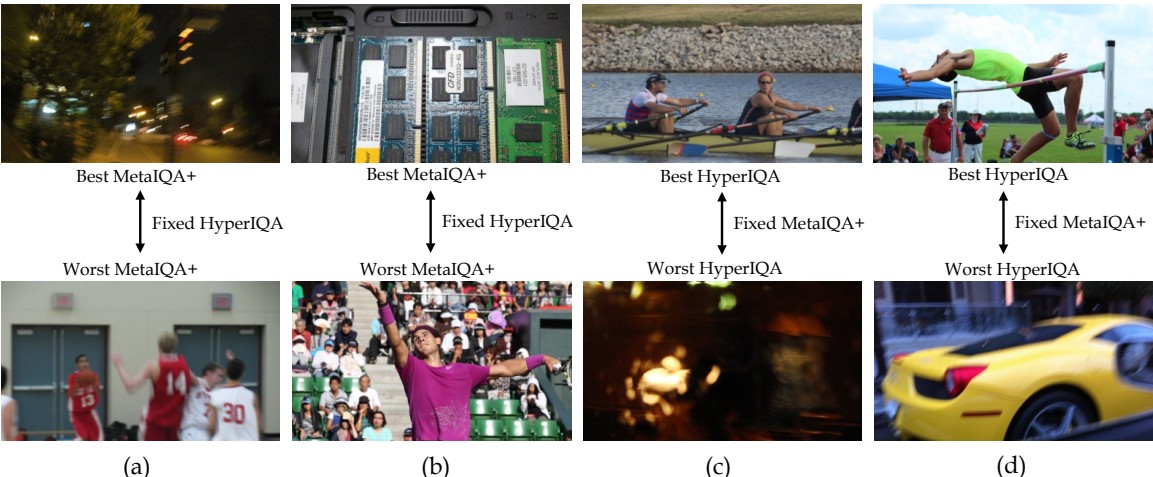

Figure 2: Representative gMAD pairs between HyperIQA and MetaIQA+. (**a**) Fixing HyperIQA at the low quality level. (**b**) Fixing HyperIQA at the high quality level. (**c**) Fixing MetaIQA+ at the low quality level. (**d**) Fixing MetaIQA+ at the high quality level.

Table 2: Ranking results of nine BIQA models. A smaller rank indicates better performance. "Distribution" in the third column means that NIMA uses the earth mover's distance to match the ground truth and predicted 1D quality distributions. "All" in the fourth column indicates that UNIQUE is trained on the combined dataset of LIVE, CSIQ, KADID-10K, BID, CLIVE, and KonIQ-10K.

| Name | Backbone | Formulation | Training Set | Time | SRCC Rank | gMAD Rank | Δ Rank |
|---|---|---|---|---|---|---|---|
| UNIQUE [8] | ResNet-34 | Ranking | All | 2021.03 | 1 | 5 | -4 |
| KonCept512 [10] | InceptionResNetV2 | Regression | KonIQ-10k | 2020.01 | 2 | 1 | 1 |
| HyperIQA [7] | ResNet-50 | Regression | KonIQ-10k | 2020.08 | 3 | 2 | 1 |
| LinearityIQA [15] | ResNeXt-101 | Regression | KonIQ-10k | 2020.10 | 4 | 3 | 1 |
| MetaIQA+ [46] | ResNet-18 | Regression | CLIVE | 2021.04 | 5 | 8 | -3 |
| Fang2020 [21] | ResNet-50 | Regression | SPAQ | 2020.08 | 6 | 9 | -3 |
| NIMA [6] | VGG-16 | Distribution | AVA | 2018.04 | 7 | 4 | 3 |
| DeepIQA [36] | VGG-like CNN | Regression | LIVE | 2018.01 | 8 | 7 | 1 |
| RankIQA [5] | VGG-16 | Ranking | LIVE | 2017.12 | 9 | 6 | 3 |

**Results**. Table 2 compares the ranking results of the nine BIQA models in the gMAD competition and in terms of the average SRCC on the four full datasets, BID [19], CLIVE [20], KonIQ-10k [10], and SPAQ [21]. The primary observation is that the latest published models, such as UNIQUE [8], MetaIQA+ [46], and Fang2020 [21] tend to overfit the peculiarities of the training sets with advanced optimization techniques. In particular, UNIQUE learns to rank image pairs from all available datasets, MetaIQA+ adopts deep meta-learning for unseen distortion generalization, while Fang2020 enables multitask learning for incorporation of auxiliary quality-relevant information. They rank much higher in terms of average SRCC than in gMAD.

Compared to improving upon optimization techniques, selecting computational structures with more capacity as the backbones seems to be a wiser choice, as evidenced by KonCept512, HyperIQA, and LinearityIQA with high rankings in gMAD. Figure 2 shows a visual comparison of the representative gMAD pairs between MetaIQA+ based on ResNet-18 and HyperIQA based on ResNet-50. Pairs of images in (a) and (b) have similar quality according to human perception, which is consistent with HyperIQA. When the roles of HyperIQA and MetaIQA+ are reversed, it is clear that the pairs of images in (c) and (d) exhibit substantially different quality. HyperIQA correctly predicts top images to have much better quality than bottom images, and meanwhile, the weaknesses of MetaIQA+ in handling dark and blurry scenes have also been exposed.

## 3.2. Easy Dataset Problem

In order to reveal the easy dataset problem, it suffices to empirically show that the newly created datasets are less effective in falsifying current BIQA models.

**Experimental Setup**. We work with the same four datasets - BID [19], CLIVE [20], KonIQ-10k [10], and SPAQ [21]. To achieve our goal, we select a state-of-the-art BIQA model - UNIQUE [8] - that permits training on multiple datasets. We train three UNIQUEs (*i.e.*, UNIQUEv1, UNIQUEv2, and UNIQUEv3, respectively) on the combination of available datasets in chronological order. For each training setting, we randomly sample 80% images from each dataset to construct the training set, leaving the remaining 10% for validation and 10% for testing. To reduce the bias caused by the randomness in dataset splitting, we repeat the training procedure ten times, and report the median SRCC results for UNIQUE variants.

**Results**. Table 3 lists the SRCC results between predictions of UNIQUEs and MOSs of different IQA datasets as test sets. The primary observation is that as more datasets are available for training, the newly created datasets are more difficult to challenge the most recently trained UNIQUE version. For example, trained on the combination of BID, CLIVE, and KonIQ-10k, UNIQUEv3 achieves a satisfac-

Table 3: SRCC between predictions of UNIQUEs and MOSs of different test sets. "—" means that the corresponding dataset is used for joint training.

| SRCC | UNIQUEv1 | UNIQUEv2 | UNIQUEv3 |
|---|---|---|---|
| CLIVE [20] | 0.6998 | — | — |
| KonIQ-10k [10] | 0.6917 | 0.7251 | — |
| SPAQ [21] | 0.7204 | 0.7932 | 0.8112 |

tory SRCC of 0.8112 on SPAQ, which is higher than 0.7932 and 0.7204 for UNIQUEv2 and UNIQUEv1 trained with fewer data. Although SPAQ is the latest dataset, it is easier than CLIVE and KonIQ-10k, which is supported by the highest correlations obtained by UNIQUEv1 and UNIQUEv2 on SPAQ. These results are consistent with the observations in Sec. 3.1, where models trained on KonIQ-10k and CLIVE rank higher than models trained on SPAQ. What is worse, the most difficult examples (as measured by the mean squared error (MSE) between model predictions and MOSs) often share similar visual appearances (see visual examples in Figure 3). This shows that the sample diversity and difficulty of existing datasets may not be well imposed in a principled way.

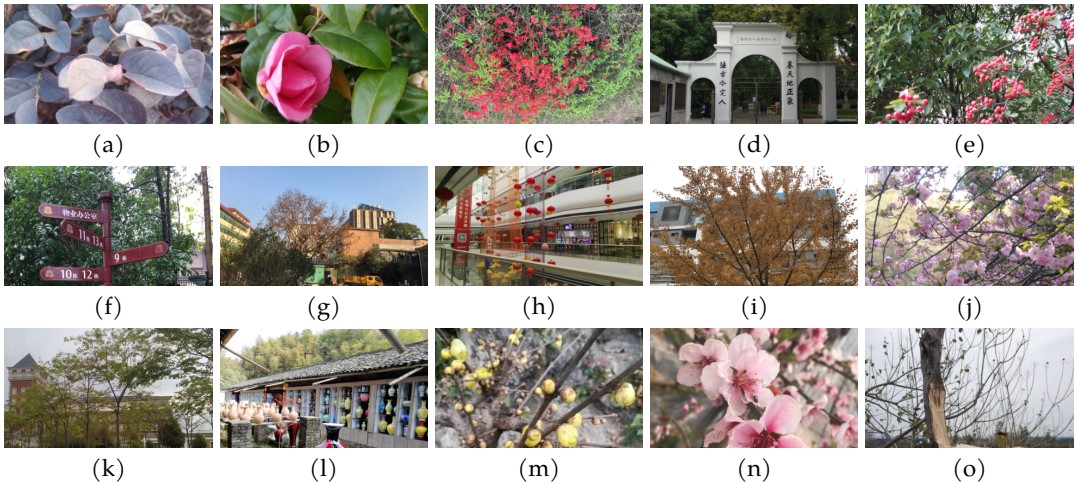

Figure 3: The top-5 difficult samples in SPAQ, as measured by the MSE between model predictions and MOSs: **(a)**-**(e)** for UNIQUEv1. **(f)**-**(j)** for UNIQUEv2. **(k)**-**(o)** for UNIQUEv3.

## 4. Integrating Model-Centric and Data-Centric IQA

In this section, we describe a computational framework for integrating model-centric and data-centric IQA approaches and provide a specific instance within the framework to alleviate the overfitting and easy dataset problems.

### 4.1. Proposed Framework

As shown in Figure 1, there is a rich body of work on how to train IQA models on available human-rated datasets, *i.e.*, the connection from data-centric IQA to model-centric IQA. The missing part

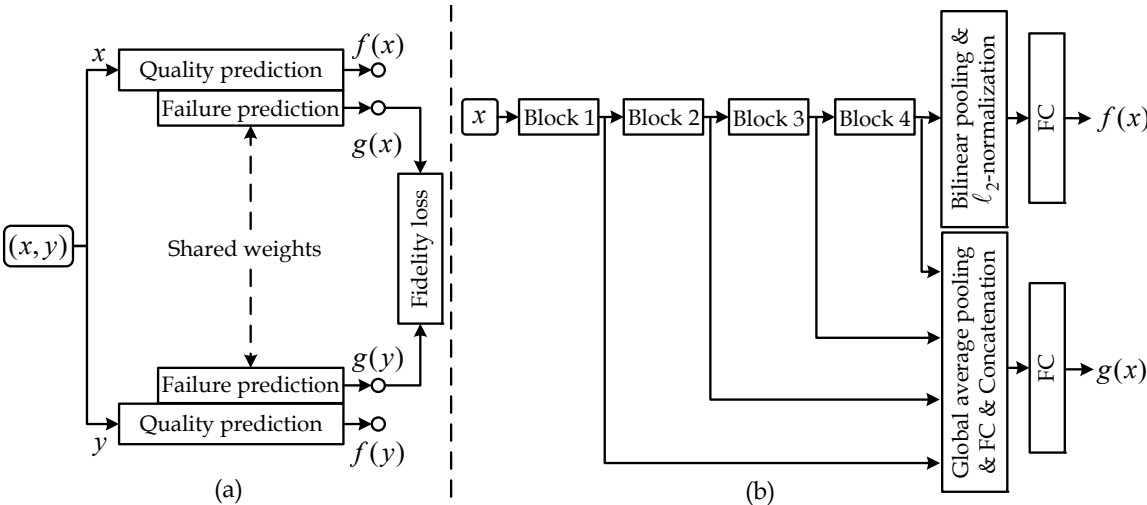

Figure 4: **(a)** The main quality predictor, $f(\cdot)$, is fixed and the auxiliary failure predictor, $g(\cdot)$, is optimized by minimizing the fidelity loss. **(b)** The backbone of $f(\cdot)$ is ResNet-34, composed of four residual blocks.

for closing the loop is to leverage existing IQA models to guide the creation of new IQA datasets. Assuming that a subjective testing environment exists, in which reliable MOSs can be collected, the problem reduces to how to sample, from a large-scale unlabeled dataset $\mathcal{U}$ with great scene complexities and visual distortions, a subset $\mathcal{D}$, whose size is constrained by the human labeling budget. Motivated by the experimental results in Sec. 3, we argue that the images in $\mathcal{D}$ are *sampling-worthy* if they are *difficult* and *diverse*. Mathematically, sample selection corresponds to the following optimization problem:

$$\mathcal{D} = \arg\max_{\mathcal{S} \subset \mathcal{U}} \mathrm{Diff}(\mathcal{S}; f) + \lambda \mathrm{Div}(\mathcal{S}), \tag{5}$$

where $\mathrm{Diff}(\cdot)$ is a difficulty measure of $\mathcal{S}$ w.r.t. the IQA model, $f(\cdot)$. It is straightforward to define $\mathrm{Diff}(\cdot)$ on a set of IQA algorithms as well. $\mathrm{Div}(\cdot)$ quantifies the diversity of $\mathcal{S}$. $\lambda$ is a trade-off parameter for the two terms. As a specific case of subset selection [49, 50], Problem (5) is generally NP-hard unless special properties of $\mathrm{Diff}(\cdot)$ and $\mathrm{Div}(\cdot)$ can be exploited. Popular approximate solutions to subset selection include greedy algorithms and convex relaxation methods.

Once $\mathcal{D}$ is identified, we collect the MOS for each $x \in \mathcal{D}$ in the assumed subjective testing environment, which makes the connection from model-centric IQA to data-centric IQA. The newly labeled $\mathcal{D}$, by construction, exposes different failures of the IQA model $f(\cdot)$, which is useful for improving its generalization. We then iterate the process of model rectification, sample selection, and subjective testing, with the ultimate goal of improving learning-based IQA from both model and data perspectives.

## 4.2. A Specific Instance in BIQA

In this subsection, we provide a specific instance of the proposed computational framework, and demonstrate its feasibility in integrating model-centric and data-centric BIQA.

To better contrast with the results in Sec. 3.2 and reduce subjective testing load, we use SPAQ to simulate the large-scale unlabeled dataset $\mathcal{U}$. The off-the-shelf BIQA model for demonstration is again UNIQUE [8], which trains on the combination of the full BID, CLIVE, and KonIQ-10k. The objective function used for optimization is the fidelity loss [42]. The optimization technique is a variant of stochastic gradient descent [51].

The core of our method is the instantiation of the sampling-worthiness module, which consists of two computational submodules to quantify the difficulty of a candidate set $\mathcal{S}$ w.r.t. to $f(\cdot)$ and the diversity of $\mathcal{S}$. Inspired by previous seminal work [52–55], we measure the difficulty through failure prediction. As shown in Figure 4, our failure predictor, $g(\cdot)$, has two characteristics: 1) it is an auxiliary module that incurs a small number of parameters; 2) it can either be solely trained while

holding the quality predictor $f(\cdot)$ fixed or jointly trained along with $f(\cdot)$. Recall that our goal is to expose diverse failures of existing BIQA models, and thus it is preferred not to re-train or fine-tune $f(\cdot)$. For the subsequent experiments, we choose to fix the quality predictor.

The failure predictor $g(\cdot)$ accepts the feature maps of the input image $x$ from each *fixed* residual block as inputs, and summarizes spatial information via global average pooling. Each stage of pooled features then undergoes an FC layer with the same number of output channels, $C$, followed by ReLU nonlinearity. After that, the four feature vectors of the same length are concatenated to pass through another FC layer to compute a scalar $g(x)$ as the indication of the difficulty of learning $x$. Assuming Gaussianity of $g(x)$ with unit variance, the probability that $x$ is more difficult than $y$ is calculated by

$$\hat{p}_{\mathrm{F}}(x, y) \;=\; \Phi\left(\frac{g(x) - g(y)}{\sqrt{2}}\right). \quad (6)$$

Table 4: SRCC results of the proposed sampling-worthiness module against six competing methods with and without the diversity measure. The large-scale unlabeled set $\mathcal{U}$ is simulated with SPAQ [21]. A lower SRCC in $\mathcal{D}$ indicates a stronger capability of failure identification. RD: Representativeness-diversity.

| Method | Without diversity | | With diversity | |
|---|---|---|---|---|
| | $\mathcal{D}$ | $\mathcal{U} \setminus \mathcal{D}$ | $\mathcal{D}$ | $\mathcal{U} \setminus \mathcal{D}$ |
| Random sampling | 0.8452 | 0.8382 | 0.8373 | 0.8383 |
| Sampling by RD [33] | 0.5932 | 0.8383 | 0.5575 | 0.8381 |
| UNIQUE uncertainty [8] | 0.5633 | 0.8397 | 0.5477 | 0.8400 |
| Query by committee [56] | 0.5487 | 0.8395 | 0.5352 | 0.8395 |
| Core-set selection [31] | 0.4968 | 0.8396 | 0.3796 | 0.8398 |
| MC dropout [30] | 0.4902 | 0.8376 | 0.3841 | 0.8379 |
| Proposed | **0.1894** | 0.8362 | **0.1413** | 0.8364 |

For the same training pair $(x, y)$, the ground-truth label can be computed by

$$p_{\mathrm{F}}(x, y) = \begin{cases} 1 & \text{if } |f(x) - \mu(x)| \geq |f(y) - \mu(y)|, \\ 0 & \text{otherwise,} \end{cases} \quad (7)$$

where $\mu(\cdot)$ represents the MOS. That is, $p_{\mathrm{F}}(x, y) = 1$ indicates that $x$ is more difficult to learn than $y$, as evidenced by a higher absolute error. We learn the parameters of the failure predictor (*i.e.*, five FC layers) by minimizing the fidelity loss between $p_{\mathrm{F}}(x, y)$ and $\hat{p}_{\mathrm{F}}(x, y)$:

$$\ell(x, y, p_{\mathrm{F}}) = 1 - \sqrt{p_{\mathrm{F}}(x, y)\hat{p}_{\mathrm{F}}(x, y)} - \sqrt{(1 - p_{\mathrm{F}}(x, y))(1 - \hat{p}_{\mathrm{F}}(x, y))}. \quad (8)$$

One significant advantage of the learning-to-rank formulation of failure prediction is that $g(\cdot)$ is independent of the scale of $f(\cdot)$, which may oscillate over iterations [55] if joint training is enabled. After sufficient training, we may adopt $g(\cdot)$ to quantify the difficulty of $\mathcal{S}$:

$$\mathrm{Diff}(\mathcal{S}) = \frac{1}{|\mathcal{S}|} \sum_{x \in \mathcal{S}} g(x), \quad (9)$$

where $|\mathcal{S}|$ denotes the cardinality of the set $\mathcal{S}$. We next define the diversity of $\mathcal{S}$ as the mean pairwise distances computed from the $1,000$-dim logits of the VGGNet [29]:

$$\mathrm{Div}(\mathcal{S}) = \frac{1}{|\mathcal{S}|^2} \sum_{(x, y) \in \mathcal{S}} \|\mathrm{logit}(x) - \mathrm{logit}(y)\|_2^2, \quad (10)$$

which provides a reasonable account for the semantic dissimilarity. While maximizing $\mathrm{Diff}(\mathcal{S})$ in Eq. (9) enjoys a linear complexity in the problem size, it is not the case when maximizing $\mathrm{Div}(\mathcal{S})$ [57]. To facilitate subset selection, we use a similar greedy method (in Eq. (4)) to solve Problem (5). Assuming $\mathcal{D} = \{x_{k'}^{\star}\}_{k'=1}^{k-1}$ is the (sub)-optimal subset that contains $k - 1$ images, the $k$-th optimal image can be chosen by

$$x_k^{\star} = \underset{x \in \mathcal{U} \setminus \mathcal{D}}{\arg\max} \, g(x) + \frac{\lambda}{k-1} \sum_{k'=1}^{k-1} \|\mathrm{logit}(x) - \mathrm{logit}(x_{k'}^{\star})\|_2^2. \quad (11)$$

## 4.3. Experiments

**Experimental Setup**. Training is carried out by minimizing the fidelity loss in Eq. (8) for failure prediction while fixing the quality predictor. The output channel of the four FC layers for feature projection is set to $C = 128$. All five FC layers in $g(\cdot)$ are initialized by He's method [58]. We adopt

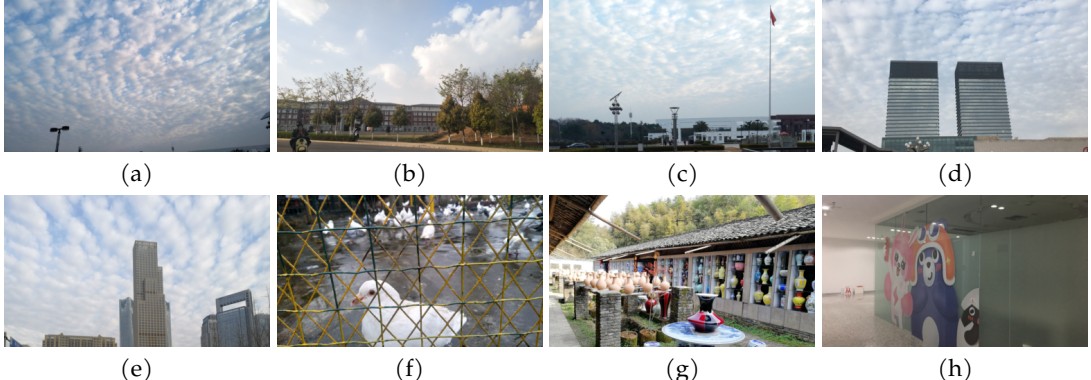

Figure 5: Representative images selected from SPAQ by the proposed sampling-worthiness module. **(a)-(d)**/**(e)-(h)** are selected images without/with the diversity measure.

Adam [51] with a mini-batch size of 32, an initial learning rate of $10^{-4}$ and a decay factor of 10 for every five epochs, and we train the failure predictor for fifteen epochs. During sample selection, we set the $\lambda$ in Eq. (5) to $10^{-6}$ in order to balance the scale difference between $\mathrm{Diff}(\cdot)$ and $\mathrm{Div}(\cdot)$. We use SPAQ to simulate $\mathcal{U}$ and select a subset $\mathcal{D}$ of size 100. Similarly, we repeat the training procedure five times and report the median results. We compare the failure identification capability of the proposed sampling-worthiness module against several deep active learning methods, including random sampling, sampling by representativeness-diversity [33], UNIQUE uncertainty [8], query by committee [56], core-set selection [31], and MC dropout [30].

**Failure Identification Results**. Table 4 shows the SRCC results between UNIQUE predictions and MOSs on the selected $\mathcal{D}$ and the remaining $\mathcal{U} \setminus \mathcal{D}$. A lower SRCC in $\mathcal{D}$ indicates better failure identification performance. We find that, for all methods except random sampling, the selected images in $\mathcal{D}$ are more difficult than the remaining ones. The proposed sampling-worthiness module delivers the best performance, identifying significantly more difficult samples. It is interesting to note that the failure identification performance of all methods, including the proposed failure predictor, can be enhanced by the incorporation of the diversity measure[2].

**Visual Results**. Figure 5 shows representative top-$K$ images selected from SPAQ by the proposed sampling-worthiness module. Without the diversity constraint, the failure predictor is inclined to select difficult images of similar visual appearances, corresponding to the same underlying failure cause. When the diverse constraint is imposed, the selected images are more diverse in content and distortion.

# 5. Conclusion and Future Work

In this paper, we first conducted computational studies to reveal the overfitting problem and the easy dataset problem rooted in the current development of BIQA. We believe these arise because of the weak connections from the model to the data. We then proposed a computational framework to integrate model-centric and data-centric IQA. We also provided a specific instance by developing a sampling-worthiness module for difficulty and diversity quantification. Our module has been proven flexible and effective in spotting diverse failures of BIQA models.

In the future, we will improve the current sampling-worthiness module by developing better difficulty and diversity measures. We may also search for more efficient discrete optimization techniques to solve the subset selection problem in the context of IQA. Moreover, we will certainly leverage the sampling-worthiness module to construct a large-scale challenging IQA dataset, with the goal of facilitating the development of more generalizable IQA models. Last, we hope the proposed computational framework will inspire researchers in related fields to rethink the exciting future directions of IQA.

---

[2]One subtlety is that each sampling strategy requires separate manual optimization of the trade-off parameter $\lambda$ due to different scales between the two terms in Eq. (5).

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
