# OpenReview forum: "Image Quality Assessment: Integrating Model-centric and Data-centric Approaches"
_CPAL.cc/2024/Conference — CPAL 2024 (Proceedings Track) Oral_

### Official Review · Reviewer_5DQU · 2023-09-13
**Simple but effective idea**

**Rating:** 6
**Confidence:** 1

**Review:**

This paper presents a new IQA method that combines the data-centric IQA and model-centric IQA. Basically, this paper is more likely an active learning method in my opinion. Using existing IQA models and VGGNet to select the most valuable data.

At first, the author demonstrates the drawbacks of the previous data-centric IQA and model-centric IQA. Data-centric suffers from the easy dataset problem and model-centric suffers from the overfitting problem. Then introduce a method that uses IQA models and  VGGNet to find the difficult and diverse samples.

The reason for the improvement of this method also seems to be intuitive. There are no obvious weaknesses in this paper to me.

I am not familiar with this field but I do buy the main idea of this paper.

---

### Official Review · Reviewer_1cbT · 2023-10-09
**Interesting study and discussions**

**Rating:** 7
**Confidence:** 3

**Review:**

This paper discusses the important problem of image quality assessment (IQA), which is essential to the evaluation of many computer vision applications. It brings forward that the isolation of model-centric and data-centric approaches impedes further progress in IQA, which is not well-addressed in previous works. The argument is supported by studies of existing IQA methods. It also proposes a novel IQA framework integrating model-centric and data-centric methods, which shows superiority compared to past works. The intuitions and discussions in this paper are interesting and motivating, and I believe they can call for more attention to this important problem as well as inspire more future research.

This paper mainly focuses on "unconditioned" evaluation of image qualities, but it'd be more interesting to also discuss "conditioned" evaluations, as they are widely used in a lot of tasks like image generation, novel view synthesis (e.g. NeRF), etc. For example, data-based metrics such as FID, KID, and model-based metrics such as PSNR, SSIM, LPIPS.

---

### Official Review · Reviewer_fMGt · 2023-10-09
**Review for Submission #3**

**Rating:** 6
**Confidence:** 2

**Review:**

**Paper Summary:**

This paper identifies the overfitting issue and the problem of simple datasets in blind IQA, possibly resulting from the isolation of the IQA model and data. In response, the paper presents a computational framework that integrates model-centric and data-centric IQA. This is achieved by enhancing the quality predictor with an auxiliary module to guide the sampling process. Its effectiveness is substantiated through a specific demonstration.

**Pros:**

1. Integrating model-centric and data-centric IQA is a promising approach and may shed light on future advancements in this field.

2. Recognizing the overfitting issue and the easy dataset problem can serve as valuable references for the community.

**Cons:**

1. A primary drawback of this work is its writing style, which may prove challenging for general readers unfamiliar with IQA. Ideally, the authors should initiate with the underlying rationales before delving into experimental details. The current presentation makes it challenging to grasp the nature of the easy dataset problem, and statements like "the newly created ones are more difficult to challenge the most recent UNIQUE" remain vague. Furthermore, certain background information and details, such as the outputs of an IQA model and the rationale behind the metrics chosen in Section 2, are essential for comprehending the paper's core content. The authors are strongly encouraged to refine their writing for broader accessibility.

2. Given that this paper's main technical contribution focuses on utilizing existing IQA models to inform the development of new IQA datasets, the phrase "integrating model-centric and data-centric IQA" might not accurately reflect the technical contribution of this work.

---

### Meta-Review · Area_Chair_LPnk · 2023-11-11

**Recommendation:** Accept (Poster)
**Confidence:** 4

**Metareview:**

The paper proposes a unifying approach that combines data-centric and model-centric methods for image quality assessment (IQA) problems. The authors should revise the paper to improve the readability. All reviewers agree to accept the paper.

---

### Decision · Program_Chairs · 2023-11-19

**Decision:**

Accept (Oral)

**Comment:**

The paper addresses the problem of image quality assessment (IQA) and introduces a computational framework that integrates model-centric and data-centric IQA,  to quantify the sampling-worthiness of candidate images based on blind IQA (BIQA) - hence clicking the data efficiency theme of CPAL. Reviewers appreciate the paper's contributions in recognizing the overfitting issue and the problem of simple datasets in blind IQA. They find the idea of integrating both approaches promising and interesting for future advancements in the field. However, concerns are raised about the writing style and clarity of the paper, suggesting that it should provide more background information and rationale for certain choices. Additionally, one reviewer suggests considering "conditioned" evaluations in addition to "unconditioned" ones. Overall, the paper's main idea and contributions are well-received by the reviewers.

The action PC chair for this paper is Atlas Wang, who made the decision after carefully reading the paper as well as the comments by all reviewers and AC. The decision is agreed by all PC chairs.